# Work stress, work motivation and their effects on job satisfaction in community health workers: a cross-sectional survey in China

Li Li,[1] Hongyan Hu,[2] Hao Zhou,[3] Changzhi He,[1] Lihua Fan,[1] Xinyan Liu,[1] Zhong Zhang,[1] Heng Li,[1] Tao Sun[1]

LL, HH and CH contributed equally.

[1]Department of Health Management, School of Public Health, Harbin Medical University, Harbin, China
[2]Research Service Office, The Second Affiliated Hospital, Harbin Medical University, Harbin, China
[3]Department of Emergency, Harbin Center for Disease Control and Prevention, Harbin, China

**Correspondence to**
Dr Li Li;
lilihmu@gmail.com

## ABSTRACT

**Objective:** It is well documented that both work stress and work motivation are key determinants of job satisfaction. The aim of this study was to examine levels of work stress and motivation and their contribution to job satisfaction among community health workers in Heilongjiang Province, China.

**Design:** Cross-sectional survey.

**Setting:** Heilongjiang Province, China.

**Participants:** The participants were 930 community health workers from six cities in Heilongjiang Province.

**Primary and secondary outcome measures:** Multistage sampling procedures were used to measure socioeconomic and demographic status, work stress, work motivation and job satisfaction. Logistic regression analysis was performed to assess key determinants of job satisfaction.

**Results:** There were significant differences in some subscales of work stress and work motivation by some of the socioeconomic characteristics. Levels of overall stress perception and scores on all five work stress subscales were higher in dissatisfied workers relative to satisfied workers. However, levels of overall motivation perception and scores on the career development, responsibility and recognition motivation subscales were higher in satisfied respondents relative to dissatisfied respondents. The main determinants of job satisfaction were occupation; age; title; income; the career development, and wages and benefits subscales of work stress; and the recognition, responsibility and financial subscales of work motivation.

**Conclusions:** The findings indicated considerable room for improvement in job satisfaction among community health workers in Heilongjiang Province in China. Healthcare managers and policymakers should take both work stress and motivation into consideration, as two subscales of work stress and one subscale of work motivation negatively influenced job satisfaction and two subscales of work motivation positively influenced job satisfaction.

## Strengths and limitations of this study

- This study is one of the first to examine the combined effects of work stress and work motivation on job satisfaction among urban community health workers in China since the implementation of health system reform.
- However, the instrument used in this study is not a commonly used international scale, there may be an inherent bias in self-report measures, and the small sample may limit the generalisability of the research findings.

## INTRODUCTION

As the basis of the three-tier health system in China, community health service institutions have played a very important role in improving access to healthcare, enhancing equity and improving health.[1] [2] In 2009, the Chinese central government promulgated a new set of health system reforms and called for the development of community health services. The state established basic public health service goals, which focused on providing health education, chronic disease management, and disease prevention services for urban and rural residents. From 2009 to 2012, the number of community health service institutions increased by 6254 and the number of visits increased by 193 949 million. Therefore, community health centres (CHCs) and those who work in them, are very important to the process of health system reform.

Heilongjiang Province is located in northeastern China and has a population of about 38.1 million people. There were 410 urban CHCs and 366 community health stations with 13 100 health workers as of 31 December 2012.[3] On average, there were 23 and 10 medical personnel in each CHC and

community health station, respectively. Since the introduction of CHCs, there have been difficulties with limited resources and insufficient and poorly trained staff. There were 5416 practitioners (including assistant practitioners) in community health institutions in Heilongjiang Province.[3] However, based on the population of the province and human resource planning ratios, there was an approximate shortfall of 30% in the number of general practitioners (5416 vs 7620) in 2012.[4] In addition, recent reforms have expanded the scope of public health services and increased workload without equivalent increases in staffing levels.[5 6]

In some CHCs, general practitioners, public health physicians and nurses have been working in teams, providing medical and basic public health services to community residents, both in the centres and during home visits. Many of these practitioners were initially hospital-based specialists, and the majority of public health physicians did not have a public health background. Therefore, to improve skills and knowledge, continuing medical education was compulsory and no fewer than 25 credit points were required per year for the title promotion. Other problems with the CHCs were lower wages and fewer title promotion opportunities relative to general hospitals. Limited resources and a shortage of skilled health workers created very tight bottlenecks in the provision of services, which led to many community health workers experiencing work-related stress and low work motivation, in addition to receiving low salaries and having restricted opportunities for promotion.[7 8] Many studies have shown that work stress and work motivation can greatly affect job satisfaction and, in turn, the quality and delivery of healthcare. However, few studies have focused on work stress and motivation and their effects on job satisfaction among Chinese community health workers since the implementation of the new health system reform policy.

Work stress is of great concern to managers, employees and other stakeholders in organisations. It is a complex phenomenon and has a multitude of definitions in a variety of theoretical models.[9] According to Lazarus and Folkman's[10] cognitive theory of stress and coping, work stress was defined as the interaction between the individual and the environment. This theory suggested that when demands from the environment exceed the available resources, the result was either stress or coping, depending on the individual's appraisal of the stressors. Karasek's[11] demand–control model assumed that psychological strain resulted from the joint effects of work demands and the degree of decision-making freedom available to workers facing the demands. The effort–reward imbalance model proposed that work stress resulted from a mismatch between high commitment and effort at work and low rewards, including salary, recognition and career promotion.[12] Nakasis and Ouzouni[13] defined work stress as the harmful physical and emotional responses that occur when job requirements do not match workers' capabilities,

resources and needs. In general, a greater imbalance between demands and individual abilities will result in greater stress.[14] Riggio[15] classified work stress into work task stress and work role stress. Cooper and Marshall's[16] model of job stress proposed that the intrinsic requirements of the job, role within the organisation, career development, organisational structure and climate, and relationships at work all contributed to work-related stress within an organisation. In our study, five subscales of work stress were identified based on this model. Existing research has recognized heavy workload, insufficient resources, work relationships, lack of professional respect, and lack of promotion opportunities as possibly the most salient work stressors for community health workers.[17–19] Long-term stress may be harmful to the health of workers themselves and may also affect community health service centres through employee dissatisfaction, burnout, poor performance or turnover intention.[20–24] Therefore, it is important to reduce work stress.

Work motivation can be defined as the degree of an individual's willingness to exert and maintain an effort towards attaining organisational goals.[25] It reflects the interactions between workers and their work environments. Nahavandi and Malekzadeh believed that motivation depends on a stable mind, aspiration or interest by the individual and can translate into action.[26] Motivation theory examined the process of motivation and explained why people at work behave the way they do in terms of efforts. Building on Vroom's[27] expectancy–valence theory of motivation, Porter and Lawler[28] proposed a model of intrinsic and extrinsic work motivation. This model suggested that intrinsic and extrinsic rewards were additive, and accounted for total job satisfaction. Intrinsic motivation refers to doing something for the inherent satisfaction involved and is highly autonomous (ie, self-regulated). In contrast, extrinsic motivation means doing something in order to obtain a separable outcome (ie, tangible or verbal rewards).[29 30] Peters identified job content and work environment, extrinsic benefits, autonomy and security, and transparency as factors in work motivation for health workers using factor analysis.[31] Patrick et al[32] and Wilbroad et al[33] developed a tool to measure health worker motivation and revealed that organisational commitment, conscientiousness, intrinsic job satisfaction, timeliness and attendance were the major determinants of higher motivation. Tribolet[34] explored the relationship between intrinsic and extrinsic motivation. Hoonakker et al[35] found that nurses appreciated challenges and opportunities for new learning and teamwork. Pool[36] explored the significant positive association between work motivation and job satisfaction, while Stringer et al[37] found that intrinsic motivation was positively associated, and extrinsic motivation negatively associated with job satisfaction.

In China, previous studies have reported that poor competency and excessive workload were key work stressors among community health workers.[7 19] Shi et al[38]

suggested that policymakers should focus on training and educational opportunities for primary care workers and consider ways to reduce workload stress and improve salaries. Hung et al[39] identified professional development, training opportunities, living environment, benefits and working conditions as the most important motivating factors for primary care providers in China. Ge analysed the relationship between work stress and job satisfaction among Chinese community health workers and reported that a degree of freedom in decision making and good workplace relationships were positive predictors of job satisfaction.[40] Chen et al[41] investigated relationships between work motivation, work stress and job satisfaction in cross-strait employees in Taiwan and mainland China.

The present study focuses on the major factors affecting work stress and motivation identified in previous research and provides an overview of community health workers' perspectives of work stress and motivation factors.[16 42–44] The purpose of this study was to assess the predictors of job satisfaction among community health workers in one Chinese province. A cross-sectional survey was conducted to measure levels of work stress, work motivation and job satisfaction. The key predictors of job satisfaction for community health workers were assessed with special attention given to work stress and motivation.

## METHODS
### Sample
A cross-sectional survey of community health workers was conducted from 1 March to 31 October 2013 in Heilongjiang Province, China. Based on the literature on community health services in China, a multistage, stratified sampling design was employed to ensure that study data were representative of the province.[7 40] First, six cities (Harbin, Qiqihar, Suihua, Jiamusi, Qitaihe and Heihe) were selected based on gross domestic product and the cities selected were matched according to the community health services they provided. Second, 15 CHCs were randomly selected from each city. On average, 22 medical personnel worked in each of the selected CHCs. Third, 60% of general practitioners, public health physicians, nurses and other health technical staff in each centre were chosen randomly, excluding those who were absent. The research team invited all selected staff members to participate in the study. The questionnaire included a cover page explaining the purposes and procedures of the study. The data were collected anonymously and the respondents completed the survey questionnaires privately to ensure confidentiality. Respondents were assured that participation in the survey was voluntary, and the return of questionnaires represented informed consent. The research staff stayed at the CHC and answered respondents' questions during the survey. Respondents were able to choose the best time to complete the questionnaire, such as when they

were not busy or their offices were quiet. Most completed questionnaires were collected on site by the investigator on the day of the visit. If some respondents did not finish that day, investigators set a date for retrieving the questionnaires. Respondents were asked to seal the completed questionnaires in individual envelopes provided by the research team. The questionnaire was relatively brief and no private personal information was collected. A total of 980 questionnaires were delivered to community health workers, all of which were returned. However, 50 (5.1%) were incomplete or blank, which left 930 valid questionnaires.

### Assessment tools
In the present study, Porter and Lawler's intrinsic and extrinsic motivation model, and Voom's expectancy–valence motivation theory were used to analyse the relationship between work motivation and job satisfaction. Lazarus and Folkman's[10] cognitive theory of stress and coping, and Karasek's[11] demand–control model were used to analyse the relationship between work stress and job satisfaction. The study instrument consisted of a self-administered questionnaire composed of four sections.

Section 1 focused on respondents' socioeconomic and demographic status.

Section 2 assessed work stress. Thirty items related to work stress were developed through intensive qualitative interviews with policymakers, healthcare managers and community health workers, a review of the literature, and an initial pilot study.[16 42] Then factor analysis, which is not discussed in this paper, yielded five subscales that comprised 26 items. The five-subscale solution accounted for 69.43% of the overall variance, and was found to be internally consistent (overall Cronbach's α=0.87). Based on Cooper and Marshall's[16] model of job stress, these five subscales of work stress were named work task and role, career development, wages and benefits, workplace relationships, and organisational structure and climate stress. They individually accounted for 16.05%, 25.10%, 12.00%, 9.08% and 7.20% of the overall variance, respectively, and the Cronbach's α within individual subscales ranged from 0.85 to 0.90. Respondents were asked to rate their perception of work stress on each item based on a five-point Likert scale: not at all stressful (1), slightly stressful (2), average (3), stressful (4) and very stressful (5). The Cronbach's α value for this study was 0.87.

Section 3 assessed work motivation. Twenty-one items were developed based on previous research, panel discussions and an initial pilot study.[43–45] Then three items were deleted and the 18 retained items were divided into four subscales by factor analysis, which is not discussed in this paper. The four-subscale solution accounted for 65.10% of the overall variance, and was found to be internally consistent (overall Cronbach's α=0.75). The subscales were renamed based on the conceptual meaning of the items and comprised: career development, recognition, responsibility and financial

motivation. They individually accounted for 21.20%, 19.40%, 14.60% and 9.90% of the overall variance, and the Cronbach's α within individual subscales ranged from 0.82 to 0.89. According to Porter and Lawler's[28] intrinsic and extrinsic motivation model, we defined career development and financial motivation as extrinsic motivation, and recognition and responsibility motivation as intrinsic motivation.[44] Respondents were asked to rate their motivation intensity on each item based on a five-point Likert scale: very weak (1), weak (2), average (3), strong (4) and very strong (5).

Section 4 assessed job satisfaction. In this study, a single-item measure was used to measure overall job satisfaction.[46] Respondents were asked to indicate their level of job satisfaction on a four-point Likert scale: strongly dissatisfied (1), dissatisfied (2), satisfied (3) and strongly satisfied (4). During the process of data analysis, strongly satisfied and satisfied were coded as 1, while strongly dissatisfied and dissatisfied were coded as 0.

## Data analysis

Survey results were analysed using SPSS V.17.0. Descriptive analyses included frequencies and percentages for categorical variables and means and SDs for continuous variables. Mean differences were examined using t tests and ANOVAs for relevant subgroups. We used logistic regression to measure the key predictors of job satisfaction because the dependent variable (job satisfaction) was a binary variable, which made linear regression unsuitable.

## RESULTS
### Socioeconomic and demographic status of respondents

The socioeconomic and demographic characteristics of the respondents are shown in table 1. A majority of participants were female (74.6%). General practitioners accounted for 36% of community health workers surveyed, followed by nurses (28.8%) and public health physicians (19.1%). In this survey, only 18.6% of the respondents had senior professional titles and less than half (40.2%) of them had a bachelor degree or higher. Only 19.6% of them had monthly incomes above 3000 CNY (approximately US$480$ in 2012). Nearly 90% of respondents worked more than 40 h/week.

### Work stress and motivation according to socioeconomic and demographic factors

The results of variance analysis and further multiple comparison t tests are shown in table 1. There were significant differences in scores for all five subscales of work stress according to occupation (p<0.01) and gender (p<0.05), with general practitioners and men showing higher levels of work stress.

Scores for the wages and benefits subscale of work stress differed significantly according to educational background (p<0.05) and income (p<0.05). Mid-level professionals reported significantly higher levels of stress

on the work task and role subscale (p<0.01) and in workplace relationships (p<0.05). Participants aged 35–44 and 45–54 years reported significantly higher levels of stress on the work task and role subscale (p<0.01).

Men had significantly higher levels of recognition and financial motivation (p<0.05). Younger workers (<25) had significantly higher levels of recognition motivation (p<0.05) and responsibility motivation (p<0.05). General practitioners had a higher level of recognition motivation (p<0.05).

There were no significant differences in any of the four work motivation subscale scores according to educational background, professional title or income.

### Levels of work stress, work motivation and job satisfaction

The mean score for overall perception of work stress was 3.11, which is slightly higher than the mid-point of 3 (table 2). The wages and benefits (3.60) subscale of work stress ranked highest, followed by work task and role (3.31), career development (2.96), organisational structure and climate (2.90), and relationships (2.75) (F=154.9, p<0.001). Statistically significant differences were noted in overall perception of stress and scores on all five work stress subscales between satisfied and dissatisfied respondents; those who were dissatisfied reported higher levels of work stress (p<0.001).

Career development motivation was rated highest, followed by financial, recognition and responsibility motivation (F=202.6, p<0.001). Levels of overall perception of work motivation and all subscales with the exception of financial motivation were significantly different between the satisfied and dissatisfied groups of respondents, and the satisfied workers had higher levels of work motivation (p<0.01).

Regarding motivation, career development was rated highest, followed by financial, recognition and responsibility motivation (F=202.6, p<0.001). Levels of overall perception of motivation and scores on all work motivation subscales, with the exception of financial motivation, differed significantly between the satisfied and dissatisfied respondents, and the satisfied workers reported higher levels of work motivation (p<0.01).

### Predictors of job satisfaction

In this study, 61.3% of respondents were satisfied with their jobs. Table 3 presents the results of a logistic regression model that examined the key predictors of job satisfaction, with special attention given to work stress and work motivation.

Only a few demographic characteristics were predictors of job satisfaction. We found that when scores on the career development and wages and benefits subscales of work stress increased by one grade, job satisfaction decreased by 32% (OR 0.68, p<0.05) and 37% (OR 0.63, p<0.01), respectively. When financial motivation increased by one grade, job satisfaction decreased by

**Table 1** Analysis of work stress and work motivation by socioeconomic and demographic status for respondents

| | N | Per cent | Work stress | | | | | Work motivation | | | |
|---|---|---|---|---|---|---|---|---|---|---|---|
| | | | Work task and role | Career development | Wages and benefits | Workplace relationships | Organisational structure and climate | Recognition | Career development | Responsibility | Financial |
| Occupation | | | | | | | | | | | |
| General practitioner | 335 | 36.0 | 3.53 | 3.17 | 3.78 | 2.90 | 3.14 | 3.61 | 4.20 | 3.44 | 4.15 |
| Public health physician | 178 | 19.1 | 3.20 | 2.89 | 3.70 | 2.63 | 2.96 | 3.57 | 4.11 | 3.67 | 4.06 |
| Nurse | 267 | 28.8 | 3.24 | 2.95 | 3.54 | 2.76 | 2.78 | 3.53 | 4.05 | 3.39 | 4.01 |
| Other | 150 | 16.1 | 3.09 | 2.79 | 3.45 | 2.65 | 2.84 | 3.59 | 4.11 | 3.40 | 4.03 |
| F | | | 6.91** | 4.97** | 3.45** | 3.05** | 6.25** | 0.66 | 2.31* | 1.96 | 0.99 |
| Sex | | | | | | | | | | | |
| Male | 236 | 25.4 | 3.44 | 3.10 | 3.77 | 2.88 | 3.12 | 3.71 | 4.18 | 3.50 | 4.19 |
| Female | 694 | 74.6 | 3.27 | 2.93 | 3.56 | 2.72 | 2.85 | 3.56 | 4.12 | 3.43 | 4.03 |
| F | | | 2.50* | 2.27* | 2.60* | 2.51* | 4.09* | 2.36* | 1.23 | 1.04 | 2.39* |
| Educational background | | | | | | | | | | | |
| High school or below | 110 | 11.8 | 3.18 | 2.90 | 3.36 | 2.81 | 2.72 | 3.57 | 4.13 | 3.42 | 4.05 |
| Junior college | 446 | 48.0 | 3.28 | 2.94 | 3.61 | 2.74 | 2.86 | 3.57 | 4.14 | 3.49 | 4.02 |
| College and above | 374 | 40.2 | 3.36 | 3.00 | 3.65 | 2.73 | 3.16 | 3.60 | 4.11 | 3.39 | 4.11 |
| F | | | 2.30 | 0.66 | 4.21* | 0.45 | 4.02* | 0.13 | 0.24 | 1.53 | 1.33 |
| Age in years | | | | | | | | | | | |
| <25 | 78 | 8.4 | 3.08 | 2.81 | 3.45 | 2.60 | 2.77 | 3.80 | 4.23 | 3.60 | 3.92 |
| 25–34 | 258 | 27.7 | 3.21 | 2.94 | 3.63 | 2.72 | 2.91 | 3.65 | 4.11 | 3.48 | 4.17 |
| 35–44 | 329 | 35.4 | 3.36 | 2.98 | 3.55 | 2.78 | 2.88 | 3.52 | 4.13 | 3.35 | 4.03 |
| 45–54 | 234 | 25.2 | 3.43 | 3.02 | 3.69 | 2.79 | 2.94 | 3.53 | 4.13 | 3.51 | 4.04 |
| ≥55 | 31 | 3.3 | 3.12 | 2.88 | 3.54 | 2.71 | 2.93 | 3.48 | 3.98 | 3.28 | 3.91 |
| F | | | 4.71** | 1.01 | 1.36 | 1.12 | 0.83 | 2.89* | 1.83 | 2.86* | 2.39 |
| Title | | | | | | | | | | | |
| Senior title | 42 | 4.5 | 3.12 | 3.11 | 3.38 | 2.69 | 2.73 | 3.37 | 3.97 | 3.55 | 3.96 |
| Vice-senior title | 131 | 14.1 | 3.32 | 2.92 | 3.65 | 2.63 | 2.93 | 3.46 | 4.05 | 3.25 | 4.03 |
| Middle title | 399 | 42.9 | 3.43 | 3.03 | 3.69 | 2.85 | 2.94 | 3.56 | 4.16 | 3.44 | 4.06 |
| Primary title | 299 | 32.2 | 3.20 | 2.93 | 3.54 | 2.72 | 2.87 | 3.62 | 4.12 | 3.49 | 4.08 |
| No title | 59 | 6.3 | 3.23 | 2.86 | 3.48 | 2.58 | 2.89 | 3.73 | 4.16 | 3.49 | 4.04 |
| F | | | 3.96** | 1.07 | 1.71 | 3.04* | 0.59 | 1.73 | 0.98 | 2.13 | 0.16 |
| Monthly income (CNY) | | | | | | | | | | | |
| <2000 | 361 | 38.9 | 3.24 | 2.95 | 3.69 | 2.76 | 2.90 | 3.61 | 4.15 | 3.49 | 4.09 |
| 2000–2999 | 386 | 41.5 | 3.32 | 2.96 | 3.61 | 2.75 | 2.88 | 3.59 | 4.13 | 3.40 | 4.06 |
| 3000–3999 | 139 | 14.9 | 3.43 | 2.97 | 3.44 | 2.68 | 2.96 | 3.52 | 4.02 | 3.43 | 3.97 |
| ≥4000 | 44 | 4.7 | 3.39 | 3.03 | 3.21 | 2.93 | 2.78 | 3.44 | 4.27 | 3.53 | 4.28 |
| F | | | 2.11 | 0.99 | 3.14* | 2.11 | 0.99 | 0.54 | 1.87 | 0.86 | 1.36 |
| Working hours (per week) | | | | | | | | | | | |
| <40 | 110 | 11.8 | 3.27 | 2.82 | 3.52 | 2.82 | 2.94 | 2.96 | 4.14 | 3.56 | 3.91 |
| 40–47 | 509 | 54.7 | 3.26 | 2.73 | 3.59 | 2.73 | 2.95 | 2.87 | 4.10 | 3.42 | 4.06 |
| 48–55 | 250 | 26.9 | 3.36 | 2.71 | 3.62 | 2.71 | 2.93 | 2.89 | 4.19 | 3.46 | 4.12 |
| ≥56 | 61 | 6.6 | 3.52 | 2.93 | 3.75 | 2.93 | 3.36 | 3.13 | 4.16 | 3.43 | 4.14 |
| F | | | 0.06 | 0.20 | 0.48 | 0.20 | 0.01* | 0.11 | 0.39 | 0.44 | 0.13 |

*p<0.05, **p<0.01.

**Table 2** Mean scores of the overall perception and subscales of work stress and work motivation with respect to the level of job satisfaction

| | Mean±SD Total (n=930) | Level of job satisfaction | | |
| --- | --- | --- | --- | --- |
| | | Satisfied (n=570, 61.3%) | Dissatisfied (n=360, 38.7%) | p Value |
| Work stress | | | | |
| Overall perception* | 3.11±0.68 | 2.95±0.68 | 3.37±0.60 | 0.000 |
| Work task and role† | 3.31±0.81 | 3.18±0.82 | 3.52±0.76 | 0.000 |
| Career development† | 2.96±0.87 | 2.79±0.85 | 3.22±0.83 | 0.000 |
| Wages and benefits† | 3.60±0.95 | 3.38±0.94 | 3.95±0.85 | 0.000 |
| Workplace relationships† | 2.75±0.79 | 2.61±0.79 | 2.96±0.74 | 0.000 |
| Organisational structure and climate† | 2.90±0.79 | 2.74±0.79 | 3.15±0.71 | 0.000 |
| Work motivation | | | | |
| Overall perception‡ | 3.80±0.55 | 3.86±0.55 | 3.70±0.55 | 0.000 |
| Career development§ | 4.13±0.57 | 4.24±0.51 | 3.95±0.62 | 0.000 |
| Recognition§ | 3.58±0.77 | 3.66±0.77 | 3.45±0.77 | 0.000 |
| Responsibility§ | 3.45±0.77 | 3.53±0.77 | 3.32±3.52 | 0.000 |
| Financial§ | 4.06±0.79 | 4.02±0.79 | 4.12±0.80 | 0.295 |

*Mean score of overall perception of work stress was calculated for each respondent by adding the value of each item of work stress and then dividing by the total number of items.
†Mean score of each subscale of work stress was calculated for each respondent by adding the value of each item of the subscale of work stress and then dividing by the total number of items.
‡Mean score of overall perception of work motivation was calculated for each respondent by adding the value of each item of work motivation and then dividing by the total number of items.
§Mean score of each subscale of work motivation was calculated for each respondent by adding the value of each item of the subscale of work motivation and then dividing by the total number of items.

28% (OR 0.72, p<0.01), and when recognition motivation and responsibility motivation increased by one grade, job satisfaction increased 1.86-fold (OR 2.86, p<0.01) and 0.36-fold (OR 1.36, p<0.05), respectively. Compared with nurses, general practitioners (OR 0.56, p<0.01) and public health physicians (OR 0.42, p<0.05) reported lower job satisfaction, while other technical staff (OR 1.89) reported higher job satisfaction. Workers with no title (OR 7.02, p<0.05) were more satisfied than workers with a senior title.

## DISCUSSION

Job satisfaction in community health workers is important for the sustainable development of basic healthcare in China, but health policymakers and managers have neglected it for a long time.[47] This study is one of the first to examine the level of work stress and work motivation and their combined effects on job satisfaction among urban community health workers in China since the implementation of health system reform.

Results indicated that the wages and benefits subscale of stress ranked highest, followed by the work task and role subscale. Similarly, previous research related to work stress found that low salary, heavy workload and few promotion opportunities were the most frequently cited workplace stressors.[48 49] Several factors may have contributed to these findings. In Heilongjiang Province, the average annual income of health service personnel in urban hospitals was 52 564 CNY (approximately US $8437) in 2012. In this study, 80.4% of the respondents'

annual incomes were below 36 000 CNY (approximately US$5778). These low salaries for community health workers increased their wages and benefits stress.[50] In addition, based on the population of the province and human resource planning ratios, there was an approximate shortfall of 30% in the number of general practitioners in 2012.[4] The recent reforms have also expanded the scope of public health services and increased workload without equivalent increases in staffing levels.[5 6]

Unfortunately, the present study found that scores on the career development, and wages and benefits subscales of work stress were negatively associated with job satisfaction. These findings were consistent with previous studies in which workers were likely to report low job satisfaction if they did not receive promotion and advancement opportunities or adequate salaries.[22 33 51]

With regard to work motivation, results showed the career development and financial subscales of work motivation ranked first and second, respectively, consistent with Hung and Hou's study, which found income, benefits and professional development were the most important motivating factors among community health workers in China.[39 52]

In this study, we defined career development and financial motivation as extrinsic motivation and recognition and responsibility motivation as intrinsic motivation based on the literature.[37 53] Results showed that the recognition and responsibility subscales of work motivation were positive predictors of job satisfaction, and financial motivation was a negative predictor. This was consistent with the 'crowding-in' effect, which proposes that intrinsic motivation increases job satisfaction, whereas

**Table 3** Logistic regression analysis for job satisfaction†

|  | OR | 95% CI |
|---|---|---|
| **Occupation (reference: nurse)** | | |
| General practitioner | 0.56** | 0.38 to 0.81 |
| Public health physician | 0.42* | 0.20 to 0.87 |
| Other technical staff | 1.89* | 1.04 to 3.44 |
| **Sex (reference: male)** | | |
| Female | 1.27 | 0.83 to 1.95 |
| **Educational background (reference: high school or below)** | | |
| Junior college | 0.76 | 0.43 to 1.34 |
| College and above | 0.75 | 0.41 to 1.40 |
| **Age in years (reference: <25)** | | |
| 25–34 | 0.60 | 0.30 to 1.21 |
| 35–44 | 1.10 | 0.51 to 2.42 |
| 45–54 | 1.04 | 0.45 to 2.35 |
| ≥55 | 8.53** | 1.86 to 39.01 |
| **Title (reference: senior title)** | | |
| Vice-senior title | 1.86 | 0.476 to 7.29 |
| Middle title | 2.57 | 0.67 to 9.78 |
| Primary title | 3.84 | 0.96 to 15.39 |
| No title | 7.02* | 1.53 to 32.12 |
| **Monthly income in CNY (reference: <2000)** | | |
| 2000–2999 | 0.50 | 0.26 to 0.98 |
| 3000–3999 | 0.99 | 0.64 to 1.52 |
| ≥4000 | 1.30 | 0.86 to 1.97 |
| **Weekly hours worked (reference: <40)** | | |
| 40–47 | 0.90 | 0.59 to 1.37 |
| 48–55 | 1.07 | 0.67 to 1.70 |
| ≥56 | 1.20 | 0.62 to 2.33 |
| **Work stress** | | |
| Work task and role | 0.98 | 0.74 to 1.300 |
| Career development | 0.68* | 0.49 to 0.94 |
| Wages and benefits | 0.63** | 0.50 to 0.79 |
| Workplace relationships | 0.80 | 0.59 to 1.09 |
| Organisational structure and climate | 0.97 | 0.71 to 1.33 |
| **Work motivation** | | |
| Career development | 1.13 | 0.85 to 1.505 |
| Recognition | 2.86** | 2.02 to 4.04 |
| Responsibility | 1.36* | 1.02 to 1.81 |
| Finance | 0.72** | 0.56 to 0.92 |

*p<0.05, **p<0.01.
†Strongly satisfied and satisfied coded as 1 versus strongly dissatisfied and dissatisfied coded as 0.

extrinsic motivation decreases job satisfaction.[54] It should be noted that in this study, the level of extrinsic motivation was higher than that of intrinsic motivation.

These findings have significant implications for managers of CHCs and policymakers in their efforts to improve workers' job satisfaction. First, policymakers should take measures to improve community health workers' salaries. In China, basic public health services are funded by the government and provided free by community health workers. If health workers are dissatisfied with their salaries, they may prefer to work for profit-making medical organisations instead of non-profit public health services. In the meantime, managers should pay staff based on their performance to increase staff enthusiasm and reduce their financial stress.

Second, policymakers should focus on appropriate promotion policies for community health workers. At present, it is difficult for community health workers to get title promotion for there are limited annual promotion quotas for CHCs in Heilongjiang Province and our study found only 18.6% of respondents had a senior professional title. Third, managers should provide and support their workers to attend training or continuing education. Fourth, managers and policymakers should take measures to inspire intrinsic motivation in workers. Becchetti *et al*[53] proposed that when workers do not work for financial incentives, they may find satisfaction irrespective of their salaries, even if the financial incentive is kept to a minimum. Therefore, managers and policymakers should introduce more incentives to encourage community health workers to work in order to gain responsibility or recognition.

As some subscales of work stress and work motivation can influence job satisfaction either positively or negatively, we examined levels of work stress and motivation according to demographic characteristics and found that policymakers and managers should pay more attention to three types of workers. The first group of workers are those aged between 35 and 54 years (35–44 and 45–54 age groups), who reported higher levels of stress on the work task and role subscale and lower levels of intrinsic motivation. Similar results have been reported elsewhere; in Qu's[55] study, community health workers in mid-level age groups were significantly more stressed than those in the youngest age group in one province of China. This could be related to workload or the difficulty and complexity of the work task, which is usually greater for 35–54-year-old workers, as they are the backbone of community health services. Men are the second group that requires attention. In our study, men's scores on all of the work stress subscales tended to be higher than women's, and men reported higher levels of financial motivation.[56] However, another Chinese study of primary health workers found no differences in financial motivation.[52] David and Srinika[57] found that women reported more stress in the financial rewards and role ambiguity subscales. The final group of workers identified as requiring attention consists of general practitioners, who experienced the highest stress according to all five work stress subscales and reported the highest career development motivation. General practitioners in CHCs face more difficult and complicated tasks and greater medical risk than other healthcare workers, and receive lower salaries and fewer promotion opportunities than their counterparts in general hospitals.

## Limitations of this study

The findings in this study should be viewed in light of four key limitations. First, this study was based on a small sample of community health workers, which may limit the generalisability of the research findings. Based on the literature on community health services in China, a multistage, stratified sampling design was employed to

ensure that study data were representative of the province.[7 40] Six sample cities were selected to account for the variability in regional per capita gross domestic product and the levels of healthcare development. Then 15 CHCs in each city were selected randomly. On average, there are 23 medical technical personnel in a CHC in Heilongjiang Province and there were approximately 22 health workers in each of the CHCs in our study. In addition, the proportions of general practitioners, public health physicians, nurses and other medical technical personnel in this study were close to the proportions found in the province as a whole.[3] Consequently, this sample was representative of Heilongjiang community health service providers, thereby enhancing the potential for generalisation of the study findings. Second, the instrument for assessing work stress and work motivation was developed from earlier study and discussed with experts, although not an international commonly used scale. Third, we used a cross-sectional survey, which may limit our ability to identify causal relationships between work stress and motivation and job satisfaction. Fourth, the questionnaires were self-administered and thus could have been affected by respondents' prevailing emotions. Therefore, the common method bias and the self-administration bias might have affected the results.

## CONCLUSION

It is important for healthcare managers to improve the job satisfaction of health workers in low-resource settings. In this study, we examined levels of work stress and motivation according to demographic characteristics and levels of job satisfaction; additionally, the key predictors of job satisfaction were identified using logistic regression analysis. The results indicated that community health workers rated wages and benefits highest among five subscales of work stress, and workers' extrinsic motivation was higher than their intrinsic motivation. The career development, and wages and benefits subscales of work stress and financial motivation were significant negative predictors of job satisfaction, while the recognition and responsibility subscales of work motivation were significant positive determinants.

Our findings suggest that there is considerable room for improvement in the job satisfaction of community health workers in Heilongjiang Province, and healthcare managers and policymakers should take both work stress and work motivation into consideration. First, they should pay more attention to three types of workers, who were aged between 35 and 54 years, male and general practitioners, as these particular groups reported higher work stress and extrinsic motivation. Second, they should take a variety of measures to reduce career development, and wage and benefits stress, as they were negative determinants of job satisfaction. Third, it is important for managers and policymakers to inspire workers' intrinsic motivation, as it can have a positive influence on job satisfaction.

**Acknowledgements** We are thankful to all the community health workers who participated in the study. We are also grateful to Yin Li, Xingsan Li, Zhuang Wang and Hongjuan Wei, who worked closely with the team to ensure the field survey was successfully implemented.

**Contributors** LL was responsible for study design, data analysis, and drafting and revising the manuscript. HH and CH were responsible for study design, data collection and data analysis. HZ and ZZ provided statistical expertise. XL, TS and HL carried out data collection and provided technical support. LF provided administrative support. All authors read and approved the final manuscript.

**Funding** This study was funded by the National Science Foundation of China (NSFC), Contract No. 71203050/G0308, Contract No. 71073034 and was supported by the Young Seed Foundation of the Public Health College of Harbin Medical University.

**Competing interests** None.

**Ethics approval** This study was approved by the Medical Ethics Committee of Harbin Medical University.

**Provenance and peer review** Not commissioned; externally peer reviewed.

**Data sharing statement** Factor analyses of work stress and work motivation are available from Li Li at lilihmu@gmail.com

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
