## [Reviewer comments · BMJ Open]

Some articles will have been accepted based in part or entirely on reviews undertaken for other BMJ Group journals. These will be reproduced where possible.

ARTICLE DETAILS

TITLE (PROVISIONAL)	Work stress, work motivation and their effects on job satisfaction for community health workers: A cross-sectional survey in China
AUTHORS	Li, Li; Hu, Hongyan; Zhou, Hao; He, Changzhi; Fan, Lihua; Liu, Xinyan; Zhang, Zhong; Li, Heng; Sun, Tao

VERSION 1 - REVIEW

REVIEWER	Qingyue Meng China Center for Health Development Studies, Peking University, China
REVIEW RETURNED	25-Mar-2014

GENERAL COMMENTS	This article covers an interesting and important topic on work stress, motivation, and job satisfaction of community health workers in China. Following comments could be helpful for revisions for making the work publishable. 1. An overview of current situation about community health workers needs to be elaborated in the introduction section for better understanding why this is an important topic and what are the gaps in knowledge and evidence in research.2. 980 community health workers from 90 community health centers were surveyed with an average of 11 health workers per health center. How many health workers in total in those 90 community health centers? This information is helpful to understand representativeness of the health workers selected. In addition, 15 community health centers from each of the 6 cities were selected. However, some big cities would have much more community health centers than other cities. Is this sampling frame reasonable?3. For examining the relationship of work stress, motivation, and job satisfaction, a theoretical framework is required in the method section.4. In the discussion section, the major findings were not explained connecting with the China's contexts of community health centers and workers. International studies cited are context-specific that may not appropriately used for direct comparisons with findings and policy implications from this study.5. Recommendations were made mainly for Health Managers (Community health centers). However, most of the policies including promotions and income policies are decided by the government in which the health managers can do nothings in changing the factors.
---

	6.The recently published studies on this topic were not cited.
--	--

REVIEWER	Kowalski, Christoph Institute for Medical Sociology, Health Services Research and Rehabilitation Science, Faculty of Human Science and Faculty of Medicine, University of Cologne, Germany
REVIEW RETURNED	17-Apr-2014

GENERAL COMMENTS	The paper has a straightforward research question and uses an acceptable design to address it. The research question is not new but I can see why this needs to be reported for China on a country-specific basis. The analysis and reporting of the results are fine. However, I have a couple of concerns. First, and this can hardly be ruled out: A response rate of 100% is an indication of an unethical sampling. How can not a single individual refuse to participate? Doesn't look like they had a choice. Second, I wonder why there wasn't any professional proof-reading before submission. How much is the fee for publishing in this journal? BPS 1,500? Proof-reading is less than 100 and would make it much easier for reviewers to understand the details. Third, the references are relatively dated and/or they just don't fit where they were referred to. Re-writing of the background section is necessary. Sentences like "Kazufumi et al. identified major work stress factors in an organization" raise a lot of questions, for example "which ones" or "how does this statement fit in the paper". Then, Cooper & Marshall are referred to after this statement. Overall, the literature review is not sufficient. I would also recommend to provide some more details on the instrument development. Stats: It is common practice to perform a log reg in situations like these but explanation why this procedure was chosen would be helpful. Furthermore, why report on all coefficients in the model? A more parsimonious would clearly help the reader. Discussion: I wonder how responsible the managers are for career development and wages/benefits. If their influence is limited, than the discussion should be rewritten. The authors should report the response rate based on all employees working on the sites (present workers/all employed) and comment on the representativeness of their sample. The other comments I made were only like the tip of the iceberg. The paper may reach the bar for publication in the end, but they will have a hard time justifying their conceptualization of "stress" and "work motivation". So they must at least be clear with their introduction/rationale. Btw: Table 1 was incomplete in my PDF.
---

VERSION 1 – AUTHOR RESPONSE

Reviewer: Qingyue Meng

1. An overview of current situation about community health workers needs to be elaborated in the introduction section for better understanding why this is an important topic and what are the gaps in knowledge and evidence in research.

Answer: Thank you for your valuable advice. We have provided more detailed information on the current situation about community health workers in the introduction section. Please refer to Page 4 and 5, Introduction Section, Paragraph 1, 2 and 3.

2. 980 community health workers from 90 community health centers were surveyed with an average of 11 health workers per health center. How many health workers in total in those 90 community health centers? This information is helpful to understand representativeness of the health workers selected. In addition, 15 community health centers from each of the 6 cities were selected. However, some big cities would have much more community health centers than other cities. Is this sampling frame reasonable?

Answer: Thank you for your valuable advice and your question; we have added some information on the sample section.

One of limitations of this study was that it was based on a small sample of community health workers, which may limit the generalizability of the research findings. We have made a more explicit description on the small sample in the limitations section.

Based on some precious studies about community health services in China, a multistage, stratified sampling design was employed to ensure that study data were provincially representative. On average, there are 23 medical technical personnel in a community health center in Heilongjiang province and there were approximately 22 health workers in each of the community health centers in our study. The research team visited the selected community health centers and chose 60% of general practitioners, public health physicians, nurses and other health technical staff in each center randomly, with the exception of those who were absent.

It is true that some big cities would have much more community health centers than other cities. At the same time, however, community health centers in big cities have more health workers than those of small cities. In this study, respondents came from big cities were more than those came from small cities. And the proportions of general practitioners, public health physicians, nurses, and other medical technical personnel in this study were close to the proportions found in the province as a whole.

Consequently, this sample may be representative of Heilongjiang community health service providers, thereby enhancing the potential for generalization of the study findings.

Please refer to 9 and 10, Sample Section, and Page 22, Limitation Section.

3. For examining the relationship of work stress, motivation, and job satisfaction, a theoretical framework is required in the method section.

Answer: Thanks for your valuable advice; it will make my paper more logical. In line with your suggestion, we have provided information on the theoretical framework. In the present study, Porter and Lawler's intrinsic and extrinsic motivation model, and Vroom's expectancy-valence motivation theory were used to analyze the relationship between work motivation and job satisfaction. Lazarus and Folkman's cognitive theory of stress and coping, and Karasek's demand-control model were used to analyze the relationship between work stress

and job satisfaction.

In addition, these theories were explicitly explained in the introduction section. Please refer to Page 6,7 and 8, Paragraph 4 and 5 of Introduction Section.

4. The discussion section, the major findings were not explained connecting with the China's contexts of community health centers and workers. International studies cited are context-specific that may not appropriately used for direct comparisons with findings and policy implications from this study.

Answer: Thank you for your advice, which is valuable in improving the quality of our manuscript.

We have rewritten the discussion section and explained the major findings connecting with the China's contexts of community health centers and workers. Please refer to Page 18 to 21, Paragraph 2, 3, 4,5 and 7 of Discussion Section in red.

5. Recommendations were made mainly for Health Managers (Community health centers). However, most of the policies including promotions and income policies are decided by the government in which the health managers can do nothings in changing the factors.

Answer: Thanks for your enlightening points, and it will make my recommendations on policy issues more logical. We have partly rewritten the recommendations in line with your suggestions. The revised recommendations are as follows:

These findings have significant implications for managers of community health centers and policymakers in their efforts to improve workers' job satisfaction. First, policymakers should take measures to improve community health workers' salaries. In China, basic public health services are funded by the government and provided by community health workers without cost to residents. If health workers are dissatisfied with their salaries, they may prefer to work for profit-making medical services instead of nonprofit public health services. In the meanwhile, managers should implement appropriate performance salary distribution system to arouse the enthusiasms of the staff and reduce their financial stress. Second, policymakers should focus on appropriate promotion policies for community health workers. At present, it was difficult for community health workers to get title promotion, for there were limit promotion quotas for CHCs every year in Heilongjiang Province and our study found only 18.6% of respondents had senior professional title. Third, the managers should provide and support their workers to attend training or continuing education. Fourth, managers and policymakers should take measures to inspire intrinsic motivation in workers. Becchetti proposed that when workers do not work for financial incentives, they may find satisfaction irrespective of their salaries, even if the financial incentive is kept to a minimum, and may therefore be satisfied with their jobs.⁵³ Therefore, managers and policymakers should introduce more incentives to encourage community health workers to work for responsibility or recognition.

6. The recently published studies on this topic were not cited.

Answer: The recently published studies on this topic were cited in the revised version of the manuscript.

Reviewer: C Kowalski

1. First, and this can hardly be ruled out: A response rate of 100% is an indication of an unethical sampling. How can not a single individual refuse to participate? Doesn't look like they had a choice.

Answer: Thanks for your comment and your question. We have provided more detailed information on the sample section in the revised manuscript.

In the process of survey, respondents were assured that participation in the survey was voluntary.

In this study, there were 980 questionnaires delivered to community health workers, all of which were returned. In the cover page of the questionnaire, respondents were told that the return of the questionnaire represented informed consent, therefore we reported a 100% response rate in last submitted version of manuscript. However, 50 (5.1%) of the questionnaires were incomplete or even blank, which left 930 valid questionnaires.

Although all of these 980 questionnaires were returned, which represented no one refused to participate in this survey, those blank questionnaires might indicate those respondents were reluctant to take part in this survey but did not refuse directly.

The reasons why we obtained a high response rate might be as follows:

First, the questionnaire was relatively brief and no private personal information was collected. And respondents were able to choose the best time to complete the questionnaire, such as when they were not busy or their offices were quiet.

Second, the privacy of the respondents was protected in the process of survey. The data were collected anonymously and the respondents completed the survey questionnaires privately to ensure confidentiality. Respondents were asked to seal the completed questionnaires into individual envelopes provided by the research team and returned to the research team.

2. I wonder why there wasn't any professional proof-reading before submission. How much is the fee for publishing in this journal? BPS 1,500? Proof-reading is less than 100 and would make it much easier for reviewers to understand the details.

Answer: Sorry for not having any professional proof-reading before the last submission. The revised manuscript has been edited and proofread by an editing company. I hope it will be helpful.

3. The references are relatively dated and/or they just don't fit where they were referred to. Re-writing of the background section is necessary. Sentences like "Kazufumi et al. identified major work stress factors in an organization" raise a lot of questions, for example "which ones" or "how does this statement fit in the paper". Then, Cooper & Marshall are referred to after this statement. Overall, the literature review is not sufficient. The paper should justify the conceptualization of "stress" and "work motivation".

Answer: Thanks. What you suggested sounds great, which will help us to reorganize our background section and make it look more structured and logical. So in line with your suggestion, we have added the recently published studies on this topic, rewritten the background section and made efforts to justify the conceptualization of "work stress" and "work motivation" in the revised manuscript. We hope these revisions will meet with your approval. Please refer to Page 4 to 9, Introduction Section.

Sentences like “Kazufumi et al. identified major work stress factors in an organization” have been revised to “Riggio classified work stress into work task stress and work role stress.¹⁵ Cooper and Marshall’s model of job stress proposed that intrinsic requirements of the job, role within the organization, career development, organizational structure and climate, and relationships at work constituted the domain of work-related stress within an organization.¹⁶ In our study, five subscales of work stress were named based on Cooper and Marshall’s model.”

4. I would also recommend to provide some more details on the instrument development.

Answer: According to your valuable advice, we have provided more detailed information on the instrument development. Please refer to Page 11 to 13, Assessment tools Section, Paragraph 3 and 4.

5. Stats: It is common practice to perform a log reg in situations like these but explanation why this procedure was chosen would be helpful. Furthermore, why report on all coefficients in the model? A more parsimonious would clearly help the reader.

Answer: We used logistic regression to measure the key predictors of job satisfaction because the dependent variable (job satisfaction) was a binary variable, which made linear regression unsuitable. During the process of data analysis, strongly satisfied and satisfied were coded as 1, while strongly dissatisfied and dissatisfied were coded as 0.

According to your advice, only Odds Ratio and 95CI were reported in the revised version. Please refer to Page 28, Table 3 in red.

6. Discussion: I wonder how responsible the managers are for career development and wages/benefits. If their influence is limited, than the discussion should be rewritten.

Answer: Thanks for your enlightening points, which will make my recommendations on policy issues more logical. In line with your suggestion, we have rewritten the discussion section and re-summarized managers and policymakers’ responsibilities in career development and wages/benefits. Please refer to Page 20, Paragraph 6 of Discussion Section.

In China, it is the policymakers who are primarily responsible for making the career development policies for community health workers. And the managers of community health centers are responsible for supporting and providing opportunities for their workers to attend training or continuing medical education.

And the community health workers’ income includes fixed base salary and performance pay. The fixed base salary was funded by governments and the managers have the right to implement the performance salary distribution system and to decide how to appraise their workers’ performance.

Therefore, policies and systems made by policymakers and managers can influence the workers’ career development and wages/benefits.

7. Btw: Table 1 was incomplete in my PDF.

Answer: Table 1 is complete in this revised version of the manuscript in PDF. Please refer to Page 25 and 26.

All the pages indicated above are in the revised manuscript. We acknowledge the reviewer’s

comments and suggestion very much, which are valuable in improving the quality of our manuscript.

VERSION 2 – REVIEW

REVIEWER	Meng Qingyue China Center for Health Development Studies, Peking University, China
REVIEW RETURNED	17-May-2014

- The reviewer completed the checklist but made no further comments.